# Better Full-Matrix Regret via Parameter-Free Online Learning

**Ashok Cutkosky**
Department of Electrical and Computer Engineering
Boston University
Boston, Massachusetts, USA
ashok@cutkosky.com

## Abstract

We provide online convex optimization algorithms that guarantee improved full-matrix regret bounds. These algorithms extend prior work in several ways. First, we seamlessly allow for the incorporation of constraints without requiring unknown oracle-tuning for any learning rate parameters. Second, we improve the regret analysis of the full-matrix AdaGrad algorithm by suggesting a better learning rate value and showing how to tune the learning rate to this value on-the-fly. Third, all our bounds are obtained via a general framework for constructing regret bounds that depend on an arbitrary sequence of norms.

## 1  Introduction

This paper provides new algorithms for online learning, which is a popular problem formulation for modeling streaming and stochastic optimization [Zinkevich, 2003, Cesa-Bianchi and Lugosi, 2006, Shalev-Shwartz, 2011, McMahan, 2014, Hazan, 2019]. Online learning is a game of $T$ rounds between an algorithm and the environment. In each round, the algorithm first chooses a point $w_t$ in some domain $W \subset \mathbb{R}^d$, after which the environment presents the learner with a loss function $\ell_t : W \to \mathbb{R}$. Performance is measured by the *regret*, which is a function of some benchmark point $\mathring{w}$: $R_T(\mathring{w}) = \sum_{t=1}^{T} \ell_t(w_t) - \ell_t(\mathring{w})$.

In order to make the problem tractable, we will assume that each $\ell_t$ is convex and $W \subset \mathbb{R}^d$ is a convex domain, which is often called *online convex optimization*. Now, if we let $g_t$ be an arbitrary subgradient of $\ell_t$ at $w_t$, we have:

$$R_T(\mathring{w}) \leq \sum_{t=1}^{T} \langle g_t, w_t - \mathring{w} \rangle$$

Because of this fact, for the rest of this paper we consider exclusively the case of linear losses and take $\sum_{t=1}^{T} \langle g_t, w_t - \mathring{w} \rangle$ as the definition of $R_T(\mathring{w})$. Well-known lower bounds [Abernethy et al., 2008] tell us that even if the environment is restricted to $\|g_t\|_2 \leq 1$ and $\|\mathring{w}\|_2 \leq 1$, no algorithm can guarantee regret better than $O(\sqrt{T})$ in all scenarios, and this bound is in fact obtained by online gradient descent [Zinkevich, 2003]. In order to go beyond this minimax result, there is a large body of work on designing *adaptive* algorithms [Auer et al., 2002, Hazan et al., 2008, Duchi et al., 2010, McMahan and Streeter, 2010, 2012, Foster et al., 2015, Orabona, 2014, Orabona and Pál, 2016, Foster et al., 2018, Jun and Orabona, 2019, Kempka et al., 2019, van der Hoeven, 2019, Mhammedi and Koolen, 2020]. Many of these prior algorithms obtain bounds like:

$$R_T(\mathring{w}) \leq \|\mathring{w}\|_2 \sqrt{\sum_{t=1}^{T} \|g_t\|_2^2} \tag{1}$$

where $\|\cdot\|_2$ is the standard 2-norm. This type of bound is appealing: in the worst-case we never do worse than the minimax optimal rate, but in many cases we can do much better. For example, if $\|\mathring{w}\|_2$ is small (intuitively, the benchmark point is "simple"), or if the $\|g_t\|_2$ values are small (intuitively, the losses are "simple"), then we obtain low regret. The challenge in obtaining these kinds of bounds lies in the fact that the values that appear in the regret guarantee are unknown to the algorithm and so intuitively the algorithm must somehow learn about them on-the-fly.

For a more nuanced form of adaptive bound, one can look to the *full-matrix* bounds. A prototypical such bound takes the form:

$$R_T(\mathring{w}) \leq \tilde{O}\left(\sqrt{r\sum_{t=1}^{T}\langle g_t, \mathring{w}\rangle^2}\right) \tag{2}$$

where here $r$ is the rank of the subspace spanned by the $g_t$. Such a bound may be desirable because it in some sense "ignores irrelevant directions" in the $g_t$ by projecting them all along $\mathring{w}$. Such bounds are available in the prior work of Mhammedi and Koolen [2020], Kotłowski [2019], Cutkosky and Orabona [2018] for the unconstrained setting when $W = \mathbb{R}^d$. For the constrained setting, to the best of our knowledge such a bound is only available in Koren and Livni [2017] subject to tuning a learning rate optimally using prior knowledge of $\sum_{t=1}^{T}\langle g_t, \mathring{w}\rangle^2$, which is unlikely to be available. In this paper we provide the first algorithm the achieve (2) for general convex domains $W$ rather than entire vector spaces.

Next, we provide a refined analysis of the regret of the full-matrix AdaGrad algorithm [Duchi et al., 2010]. Prior analysis of full-matrix AdaGrad yields a regret bound that is never better than $(1)$[1]. Nevertheless, full-matrix AdaGrad is empirically successful, suggesting that something is missing from the analysis. We posit that the missing ingredient is a suboptimal tuning of the learning rate, and show that with oracle tuning (which is unavailable apriori) one can obtain the regret bound:

$$R_T(\mathring{w}) \leq \tilde{O}\left[\sqrt{\left\langle \mathring{w}, \sqrt{\sum_{t=1}^{T} g_t g_t^\top}\ \mathring{w}\right\rangle \mathrm{tr}\sqrt{\sum_{t=1}^{T} g_t g_t^\top}}\right] \tag{3}$$

We provide an interpretation of this bound suggesting that it allows for small regret when $\sum_{t=1}^{T} g_t g_t^\top$ is *approximately* low-rank, and develop an algorithm that obtains this bound with requiring any tuning. Intriguingly, the three regret bounds (1), (2), and (3) are all incomparable - there are sequences of $g_t$ such that any one of them might be significantly better than the others.

In order to obtain these results, we develop a general algorithm that takes a sequence of increasing norms $\|\cdot\|_0, \ldots, \|\cdot\|_{T-1}$ and obtains regret

$$R_T(\mathring{w}) \leq \tilde{O}\left(\|\mathring{w}\|_{T-1}^2\sqrt{\sum_{t=1}^{T}\|g_t\|_{t-1,\star}^2}\right) \tag{4}$$

Here $\|\cdot\|_{t,\star}$ is the dual norm to $\|\cdot\|_t$. The norms $\|\cdot\|_t$ may be generated on-the-fly (e.g. $\|\cdot\|_t$ can depend on $g_t$). Further, our algorithm can incorporate arbitrary convex domains $W$. Prior adaptive algorithms have typically required specific forms of $W$, such as being an entire vector space or having bounded diameter, and have often focused on a single norm.

This paper is organized as follows: in Section 2, we lay out our setting and introduce some background from the literature. In Section 3, we describe our our intermediate result achieving the bound (4). Then, in Sections 4 and 5, we show how to use our approach to achieve bounds (2) and (3).

## 2 Preliminaries

### 2.1 Notation and Setup

We use $\|\cdot\|_0, \ldots, \|\cdot\|_{T-1}$ to indicate an arbitrary sequence of $T$ potentially different norms. To avoid confusion between the $L_p$ norm and the $p$th element of a sequence of norms, we denote the $L_p$

norm using a bold font: $\| \cdot \|_{\mathbf{p}}$. For a symmetric positive-definite matrix $M$, the norm defined by $M$ is denoted by $\|x\|_M = \sqrt{x^\top M x}$. We will use the notation $G_t = \sum_{i=1}^{t} g_i g_i^\top$ as a shorthand for the sum of the outer product of the loss vectors $g_t$. Finally, the dual of a norm is $\|x\|_\star = \sup_{\|y\| \le 1} \langle y, x \rangle$.

We restrict our attention to those norms such that the function $\frac{1}{2} \| \cdot \|^2$ is $\sigma$-strongly-convex with respect to the same norm $\| \cdot \|$ for some $\sigma$. A function $f : W \to \mathbb{R}$ is $\sigma$-strongly-convex if for all $x$ and $y$ and $g \in \partial f$ we have $f(y) \ge f(x) + \langle g, y - x \rangle + \frac{\sigma}{2} \|x - y\|^2$.

We will assume $W$ is a convex set for which it is possible to compute the projection operation $\Pi(x) = \operatorname{argmin}_{w \in W} \|w - x\|$ for any norm $\| \cdot \|$ we are interested in. We will also usually require $\|g_t\|_\star \le 1$ for the norms we consider. We recall for convenience here that $\|g\|^2_{M,\star} = \langle g, M^{-1} g \rangle$

Finally, in order to ease exposition we have suppressed many constants and occasionally a logarithmic factor in our main presentation. We provide full characterizations of all our results including constant factors in the Appendix along with any proofs not in the main text.

In the next subsections, we describe some material from prior literature we will use to construct our algorithms.

## 2.2   Follow-the-Regularized-Leader

Follow-the-Regularized-Leader (FTRL) [Shalev-Shwartz, 2007] is one of the most successful abstractions for designing online convex optimization algorithms (see McMahan [2014] for a detailed survey). FTRL algorithms produces $w_1, \ldots, w_T$ through the use of *regularizer* functions $\psi_0, \ldots, \psi_{T-1}$. Specifically, $w_{t+1}$ is given by:

$$w_{t+1} = \operatorname*{argmin}_{w \in W} \psi_t(w) + \sum_{i=1}^{t} \langle g_i, w \rangle$$

The following result characterizes the regret of FTRL:

**Theorem 1** (Adapted from McMahan [2014] Theorem 1). *Suppose each $\psi_t$ is $\sigma_t$-strongly-convex with respect to a norm $\| \cdot \|_t$ for some $\sigma_t$, and $\psi_{t+1}(w) \ge \psi_t(w)$ for all $t$ and all $w \in W$. Further suppose $\inf_{w \in W} \psi_0(w) = 0$. Then the regret of FTRL is bounded by:*

$$R_T(\mathring{w}) \le \psi_{T-1}(\mathring{w}) + \frac{1}{2} \sum_{t=1}^{T} \frac{\|g_t\|^2_{t-1,\star}}{\sigma_{t-1}}$$

*where recall we define $\|g\|_\star = \sup_{\|x\| \le 1} \langle g, x \rangle$ for any seminorm $\| \cdot \|$.*

The FTRL algorithm template has been used to great effect through clever choices of regularizer functions $\psi_t$. However, most prior adaptive algorithms based on FTRL (e.g. [Duchi et al., 2010, McMahan and Streeter, 2010]) require tuning some learning rate parameter $\eta$ to the value of $\|\mathring{w}\|$. Unfortunately, the optimal value of $\eta$ is unknown a priori (and maybe even a posteriori) because we do not know what $\|\mathring{w}\|_{\mathbf{2}}$ is.

## 2.3   Parameter-Free Algorithms

In an effort to fix the need to tune learning rates, much work has gone into designing "parameter-free" algorithms that can adapt to unknown values of $\mathring{w}$ [McMahan and Streeter, 2012, Orabona, 2013, Orabona and Pál, 2016, Foster et al., 2017a, Cutkosky and Boahen, 2017, Foster et al., 2018, Cutkosky and Orabona, 2018, Kempka et al., 2019]. These algorithms make use of a known bound on the norm of $g_t$ in order to achieve adaptivity to $\|\mathring{w}\|$. We will make use of the following recent bound (which is optimal up to constants and quantities inside logarithms):

**Theorem 2** (Adapted from Cutkosky and Sarlos [2019] Theorem 2). *For any user-specified values $\epsilon > 0$ and $0 \le Z \le 1$, there exists an online convex optimization algorithm with domain $W = \mathbb{R}$*

*that runs in time $O(1)$ per update such that if $|g_t| \leq 1$ for all $t$, the regret is bounded by:*

$$R_T(\mathring{w}) = \sum_{t=1}^{T} g_t(w_t - \mathring{w}) \leq O\left[ \epsilon + |\mathring{w}| \max \left( \sqrt{ \frac{1 + \sum_{t=1}^{T} g_t^2}{Z} \log \left( 1 + \frac{\left( \frac{1+\sum_{t=1}^{T} g_t^2}{Z} \right)^{\frac{1}{2}+\frac{Z}{2}} |\mathring{w}|}{\epsilon} \right) }, \right.\right.$$
$$\left.\left. \log \left( 1 + \frac{\left( \frac{1+\sum_{t=1}^{T} g_t^2}{Z} \right)^{\frac{1}{2}+\frac{Z}{2}} |\mathring{w}|}{\epsilon} \right) \right) \right] \tag{5}$$

In order to ease notation in our results, we will just set $Z = 1$ and drop the $Z$ dependency in Theorem 2 from all bounds in the paper. For completeness, we provide a proof of this result in Appendix F.

## 3 Adapting to Varying Norms

In this Section, we show our how to achieve the regret bound (4) in arbitrary convex domains $W$. We decompose the problem into three stages: first, we use FTRL to obtain an regret bound with a very poor dependence on $\|\mathring{w}\|_{T-1}$. Then, we will show how to combine this with a one-dimensional parameter-free algorithm to obtain the desired bound in the case that $W$ is an entire vector space. Finally, we will show how to constrain our algorithm to arbitrary convex $W$. Although each of our individual steps is pleasingly straightforward, the final result is surprisingly powerful, as it will allow us to easily obtain our new full-matrix bounds.

Our FTRL algorithm is reminiscent of prior adaptive methods, but we enforce a special *time varying constraint*. This will make the algorithm much worse on its own, but allow for an overall improvement later. Specifically, suppose we have a sequence of norms $\|\cdot\|_0, \ldots \|\cdot\|_{T-1}$ such that $\|x\|_t \geq \|x\|_{t-1}$, and $\frac{1}{2}\|\cdot\|_t^2$ is $\sigma$-strongly-convex with respect to $\|\cdot\|_t$ for all $t$ and $x$. Consider FTRL with regularizers:

$$\psi_t(w) = \begin{cases} \frac{1}{\sqrt{\sigma}} \|w\|_t^2 \sqrt{1 + \sum_{i=1}^{t} \|g_i\|_{i-1,\star}^2} & \text{if } \|w\|_t \leq 1 \\ \infty & \text{if } \|w\|_t > 1 \end{cases} \tag{6}$$

Then we have the following corollary of Theorem 1:

**Lemma 3.** *Let $W$ be a real vector space and $\|\cdot\|_1, \ldots, \|\cdot\|_T$ are an increasing sequence of norms on $W$ such that $\frac{1}{2}\|\cdot\|_t$ is $\sigma$-strongly-convex with respect to $\|\cdot\|_t$. Suppose we run FTRL with regularizers given by (6), and with $g_t$ satisfying $\|g_t\|_{t-1,\star} \leq 1$ for all $t$. Then $\|w_t\|_{t-1} \leq 1$ for all $t$, and for all $\mathring{w}$ with $\|\mathring{w}\|_{T-1} \leq 1$, the regret of FTRL is bounded by*

$$R_T(\mathring{w}) \leq \frac{1}{\sqrt{\sigma}} \left( \|\mathring{w}\|_{T-1}^2 \sqrt{1 + \sum_{t=1}^{T-1} \|g_t\|_{t-1,\star}^2} + \sqrt{\sum_{t=1}^{T} \|g_t\|_{t-1,\star}^2} \right).$$

Note that this bound is actually much worse than one might typically expect, as the outputs are constrained to smaller and smaller balls over the course of the algorithm. Counterintuitively, this property is crucial for improved results in the next section.

### 3.1 Unconstrained Domains

Now, with Lemma 3 in hand, we will proceed to build an algorithm that achieves the bound (4) in the *unconstrained* setting. Our technique is based on the dimension-free to one-dimensional optimization reduction proposed by Cutkosky and Orabona [2018], taking into account the particular dynamics of our FTRL algorithm. The pseudocode for this technique is presented in Algorithm 1 below.

---
**Algorithm 1** Unconstrained Varying Norms Adaptivity
---
**Input:** sequence of norms $\|\cdot\|_0, \dots, \|\cdot\|_{T-1}$, real vector space $W$, strong-convexity parameter $\sigma$.

Instantiate one-dimensional parameter-free online learning algorithm $\mathcal{A}$ from Theorem 2.

Set $\psi_0(x) = \frac{1}{\sqrt{2\sigma}}\|x\|_0^2$.

Set $x_1 = \operatorname{argmin}_{w \in W} \psi_0(w)$.

**for** $t = 1 \dots T$ **do**

    Get $y_t \in \mathbb{R}$ from $\mathcal{A}$.

    Output $w_t = y_t x_t$ and get $g_t$.

    Set $\psi_t(x) = \begin{cases} \frac{1}{\sqrt{2\sigma}}\|x\|_t^2 \sqrt{1 + \sum_{i=1}^t \|g_i\|_{i-1,\star}^2} & \text{if } \|x\|_t \le 1 \\ \infty & \text{if } \|x\|_t > 1 \end{cases}$

    Set $x_{t+1} = \operatorname{argmin}_{w \in W} \psi_t(w) + \sum_{i=1}^t \langle g_i, w \rangle$.

    Send $s_t = \langle g_t, x_t \rangle$ to $\mathcal{A}$ as the $t$th loss.

**end for**
---

**Lemma 4.** *Under the assumptions of Lemma 3, for any $\mathring{w} \in W$ (recall we assume $W$ is an entire vector space in Lemma 3), the regret of Algorithm 1 is bounded by:*

$$R_T(\mathring{w}) \le O\left[\epsilon + \frac{2\|\mathring{w}\|_{T-1}}{\min(1, \sqrt{\sigma})} \max\left(\sqrt{1 + \sum_{t=1}^T \|g_t\|_{t-1,\star}^2 \log\left(1 + \frac{\sum_{t=1}^T \|g_t\|_{t-1,\star}^2 \|\mathring{w}\|_{T-1}}{\epsilon}\right)}, \right.\right.$$
$$\left.\left. \log\left(1 + \frac{\sum_{t=1}^T \|g_t\|_{t-1,\star}^2 \|\mathring{w}\|_{T-1}}{\epsilon}\right)\right)\right]$$

## 3.2 Adding Constraints

Algorithm 1 provides a method for obtaining the bound (4) when $W$ is an entire vector space, so in this section we show how to fix the algorithm so that $W$ may be an arbitrary convex domain. We do this by again appealing to a technique from [Cutkosky and Orabona, 2018]. This time, we use their Theorem 3, which provides a way to produce constrained algorithms from unconstrained algorithms. The original result considers only the case of a fixed norm and is applied to achieve bounds like (1). We modify the technique to consider varying norms as well. The algorithm is presented in Algorithm 2 below, and the analysis achieving (4) is in Theorem 5.

---
**Algorithm 2** Varying Norms Adaptivity
---
**Input:** Convex domain $W$ in a real vector space $V$.

Define $\Pi_t(v) = \operatorname{argmin}_{w \in W} \|v - w\|_{t-1}$.

Define $S_t(v) = \|v - \Pi_t(v)\|_{t-1}$.

Initialize Algorithm 1 with domain $V$ using the algorithm of Theorem 2 as the base learner.

**for** $t = 1 \dots T$ **do**

    Get $t$th output $v_t \in V$ from Algorithm 1.

    Output $w_t = \Pi_t(v_t)$, and get loss $g_t$.

    Define $\ell_t(v) = \frac{1}{2}\left(\langle g_t, v \rangle + \|g_t\|_{t-1,\star} S_t(v)\right)$.

    Let $\hat{g}_t \in \partial \ell_t(v_t)$, and send $\hat{g}_t$ to Algorithm 1 as the $t$th loss.

**end for**
---

**Theorem 5.** *Each output $w_t$ of Algorithm 2 lies in $W$, and the regret for any $\mathring{w} \in W$ is at most:*

$$R_T(\mathring{w}) \le O\left[\epsilon + \frac{\|\mathring{w}\|_{T-1}}{\min(1, \sqrt{\sigma})} \max\left(\sqrt{1 + \sum_{t=1}^T \|g_t\|_{t-1,\star}^2 \log\left(1 + \frac{\sum_{t=1}^T \|g_t\|_{t-1,\star}^2 \|\mathring{w}\|_{T-1}}{\epsilon}\right)}, \right.\right.$$
$$\left.\left. \log\left(1 + \frac{\sum_{t=1}^T \|g_t\|_{t-1,\star}^2 \|\mathring{w}\|_{T-1}}{\epsilon}\right)\right)\right]$$

## 4 Full-Matrix Bounds

The results of the previous section operate with arbitrary norms and in potentially infinite dimensional spaces. In this section and the next, we will specialize to the case $W \subset \mathbb{R}^d$, and show how to obtain so-called "full-matrix" or "preconditioned" regret bounds. In this section, we will consider the full-matrix regret bound given by (2).

Up to a factor of $\sqrt{\log(T)}$, this bound is achieved in the case where $W$ is an entire vector space by Mhammedi and Koolen [2020], and similar bounds utilizing various extra assumptions or worse log factors are obtained by Cutkosky and Orabona [2018], Kotłowski [2019], Cesa-Bianchi et al. [2005]. When $W$ is not an entire vector space, it seems harder to achieve this bound. However, some progress has been made in certain settings. For example, when $W$ is the probability simplex, Foster et al. [2017b] achieves a bound $\sqrt{rT}$, which adapts automatically to $r$. For more general $W$, Koren and Livni [2017] achieves the desired result if their algorithm is tuned with oracle knowledge of $\sum_{t=1}^{T} \langle g_t, \mathring{w} \rangle^2$.

To appreciate some of the subtleties of incorporating arbitrary constraints $W$, let us consider one straw-man solution. We might be tempted to start with the algorithm of Mhammedi and Koolen [2020] that achieves (2) in the unconstrained setting, and then try to add constraints through direct application of the unconstrained-to-constrained reduction proposed in Cutkosky and Orabona [2018], which is also the key component of our Algorithm 2. Unfortunately, this might fail because the alterations to the gradients necessary to incorporate the constraints may destroy the original regret bound. If the original losses are $g_t$, and the losses supplied to the unconstrained algorithm after performing the unconstrained-to-constrained transformation are $\tilde{g}_t$, the final regret will be $\tilde{O}\left(\sqrt{\tilde{r} \sum_{t=1}^{T} \langle \tilde{g}_t, \mathring{w} \rangle^2}\right)$, where $\tilde{r}$ is the rank of the $\tilde{g}_t$. It is not clear what the relationship is between this quantity and the desired bound $\tilde{O}\left(\sqrt{r \sum_{t=1}^{T} \langle g_t, \mathring{w} \rangle^2}\right)$. The reduction allows us to guarantee $\|\tilde{g}_t\|_\star \leq 2\|g_t\|_\star$ for any given norm $\|\cdot\|$, but this does not sufficiently elucidate the problem. For example, even if we knew the norm $\|\cdot\|_{G_T}$ in advance and ensured $\|\tilde{g}_t\|_{G_T^{-1}} \leq 2\|g_t\|_{G_T^{-1}}$, applying Cauchy-Schwarz would yield a regret of $\tilde{O}\left(\|\mathring{w}\|_{G_T}\sqrt{\tilde{r}\sum_{t=1}^{T}\|g_t\|_{G_T^{-1}}^2}\right) = \tilde{O}(r\|\mathring{w}\|_{G_T})$, which has the wrong dependence on the rank $r$.

Perhaps surprisingly given this difficulty, a straightforward application of Theorem 5 allows us to obtain (2), up to a factor of $\log(T)$. Note that this is $\sqrt{\log(T)}$ worse than Cutkosky and Orabona [2018], but we are able to handle any convex domain. Intuitively, one can view our Algorithm 1 as providing a specially-treated unconstrained regret bound that is designed specifically to work around the difficulties in naively applying the unconstrained-to-constrained reduction to an arbitrary unconstrained algorithm.

The key idea in our approach is that the norms $\|\cdot\|_t$ used by Algorithm 2 need not be specified ahead of time: so long as $\|\cdot\|_t$ depends only on $g_1, \ldots, g_t$, it is still possible to run the algorithm. Next, observe that $\sum_{t=1}^{T} \langle g_t, \mathring{w} \rangle^2$ can be viewed as $\|\mathring{w}\|_{G_T}^2$, where we recall that $\|\cdot\|_{G_T}$ is the norm induced by $G_T$: $\|x\|_{G_T}^2 = x^\top G_T x$. Inspired by these observations, our approach is to run Algorithm 2 using norms $\|\cdot\|_t = \|\cdot\|_{G_t}$. The algorithm is analyzed in Theorem 6 below.

**Theorem 6.** *Suppose $g_t$ satisfies $\|g_t\| \leq 1$ for all $t$ where $\|\cdot\|$ is any norm such that $\frac{1}{2}\|\cdot\|^2$ is $\sigma$-strongly convex with respect to $\|\cdot\|$. Let $G_t = \sum_{i=1}^{t} g_i g_i^\top$ and let $r$ be the rank of $G_T$. Suppose we run Algorithm 2 with $\|x\|_t^2 = \|x\|^2 + x^\top(I + G_t)x$, where $I$ is the identity matrix. Then we obtain regret $R_T(\mathring{w})$ bounded by:*

$$O\left[\sqrt{\frac{\|\mathring{w}\|^2 + \|\mathring{w}\|_2^2 + \sum_{t=1}^{T}\langle g_t, \mathring{w}\rangle^2}{\min(\sigma, 1)}} \max\left(\log\left(1 + \frac{r\log(T)\sqrt{\|\mathring{w}\|^2 + \|\mathring{w}\|_2^2 + \sum_{t=1}^{T}\langle g_t, \mathring{w}\rangle^2}}{\epsilon}\right),\right.\right.$$

$$\left.\left.\sqrt{r\log(T)\log\left(1 + \frac{r\log(T)\sqrt{\|\mathring{w}\|^2 + \|\mathring{w}\|_2^2 + \sum_{t=1}^{T}\langle g_t, \mathring{w}\rangle^2}}{\epsilon}\right)}\right) + \epsilon\right]$$

*Proof.* We have $\|x\|_t^2 = \|x\|^2 + x^\top(I + G_t)x = \|x\|^2 + \|x\|_{\mathbf{2}}^2 + \sum_{i=1}^t \langle g_t, x \rangle^2$, so that $\|\cdot\|_t$ is increasing in $t$. Further, since $\|x\|_{t-1} \geq \|x\|$, we must have $\|g_t\|_{t-1,\star} \leq 1$ for all $t$. Next, observe that since $\|g_t\|_\star \leq 1$, we have

$$\|x\|^2 + \|x\|_{\mathbf{2}}^2 + \sum_{i=1}^{t-1} \langle g_i, x \rangle^2 \geq \|x\|_{\mathbf{2}}^2 + \sum_{i=1}^t \langle g_i, x \rangle^2 = x^\top(I + G_t)x$$

Therefore, we have $\|g_t\|_{t-1,\star} \leq g_t^\top(I + G_t)^{-1}g_t$. Now recall that for any PSD matrix $M$, $\frac{1}{2}x^\top Mx$ is 1-strongly convex with respect to the norm $\sqrt{x^\top Mx}$. Therefore, by Lemma 8 (provided in supplement), we have that $\frac{1}{2}\|x\|_t^2$ is $\min(\sigma, 1)$-strongly convex with respect to $\|\cdot\|_t$ so that we have satisfied all the hypotheses of Theorem 5. Finally, before we apply Theorem 5, we need to analyze

$$\sum_{t=1}^T \|g_t\|_{t-1,\star}^2 \leq \sum_{t=1}^T g_t^\top(I + G_t)^{-1}g_t \leq \log\left(\frac{\det(I + \sum_{i=1}^t g_t g_t^\top)}{\det(I)}\right)$$
$$\leq \operatorname{rank}(G_T)\log(T+1)$$

where we have applied Lemma 11 of Hazan et al. [2007]. The result now follows from Theorem 5. $\qquad\square$

Note that for concreteness, if we set $\|\cdot\| = \|\cdot\|_{\mathbf{2}}$ in the above bound, then the norms $\|\cdot\|_t$ become the familiar matrix-based norm $\|x\|_t = \sqrt{x^\top(2I + G_t)x}$. We have opted to leave the more general formulation in place to allow for $g_t$ that are not bounded in the $L_2$ norm.

## 5  Full-Matrix Adagrad with Oracle Tuning

In this section we consider a different kind of full-matrix bound inspired by the full-matrix AdaGrad algorithm [Duchi et al., 2010]. Full-matrix AdaGrad can be described as FTRL using regularizers:[2]

$$\psi_t(x) = \frac{1}{\eta}\langle x, (I + G_t)^{1/2}, x \rangle$$

where $\eta$ is a scalar learning rate parameter that must be set by the user. $(I + G_t)^{1/2}$ indicates the symmetric positive-definite matrix square-root of $I + G_t$, which exists since $I + G_t$ is a symmetric positive-definite matrix. This algorithm is empirically very successful, in spite of the computational overhead coming from manipulating the $d \times d$ matrix $G_t$. Indeed, much work has gone into providing approximate versions of this algorithm that reduce the computation load while still retaining some performance benefits [Gupta et al., 2018, Agarwal et al., 2019, Chen et al., 2019]. Prior analyses of full-matrix AdaGrad considers domains $W$ with finite diameter $D = \sup_{x,y \in W} \|x - y\|_{\mathbf{2}}$, and suggests setting $\eta = O(D)$ to obtain a regret bound of:

$$R_T(\mathring{w}) \leq O(D\operatorname{tr}(G_T^{1/2}))$$

However, by linearity of trace and concavity of square root, we have:

$$D\operatorname{tr}(G_T^{1/2}) \geq D\sqrt{\operatorname{tr}(G_T)} = D\sqrt{\sum_{t=1}^T \|g_t\|_{\mathbf{2}}^2}$$

The bound $R_T(\mathring{w}) \leq D\sqrt{\sum_{t=1}^T \|g_t\|_{\mathbf{2}}^2}$ can be achieved by simple (and fast) online gradient descent with a scalar learning rate, $w_{t+1} = w_t - \frac{Dg_t}{\sqrt{\sum_{i=1}^t \|g_t\|_{\mathbf{2}}^2}}$, so the prior regret bound of full-matrix AdaGrad does not appear to show any benefit gained by the extra matrix computations. This poses a mystery: since the actual algorithm is so effective, it seems we are missing something in the analysis. We propose a possible explanation for this quandary. The main idea is that, in practice, the theoretical guidance to set $\eta = O(D)$ is rarely used. Instead, $\eta$ is tuned via manually checking different values to find which is empirically best. Thus, if we could show that full-matrix AdaGrad achieves gains with an oracle-tuning for $\eta$, this might explain the improved performance in practice.

To this end, recall that from Theorem 1 we can write the regret of full-matrix AdaGrad as:

$$R_T(\mathring{w}) \leq O\left(\frac{\mathring{w}^\top (I + G_T)^{1/2}\mathring{w}}{\eta} + \eta \sum_{t=1}^T g_t^\top (I + G_{t-1})^{-1/2} g_t\right) \leq O\left(\frac{\mathring{w}^\top G_T^{1/2}\mathring{w}}{\eta} + \eta \mathrm{tr}(G_T^{1/2})\right)$$

where the second inequality is due to Lemma 10 of Duchi et al. [2010], and we have ignored the dependence on $I$ for simpler exposition. Then it is clear that with the optimal tuning of $\eta = O\left(\sqrt{\frac{\langle \mathring{w}, G_T^{1/2}\mathring{w}\rangle}{\mathrm{tr}(G_T^{1/2})}}\right)$, we obtain regret bound of (3). In order to appreciate the potential of this bound, let us construct a particular sequence of $g_t$s and evaluate the bound. We will compare the bound (3) to both (2) and (1). Our example will illustrate that (3) can in some sense adapt to the case that $G_T$ is full-rank but "approximately low rank", while the analysis of the full-matrix algorithm in Section 4 does not obviously allow for such behavior.

Let $v_1, \dots, v_d$ be an orthonormal basis for the $d$-dimensional vector space containing $W$. Assume $d$ is a perfect square and $T = 2d + 2k\sqrt{d}$ for some integer $k$. For the first $d$ rounds, $g_t = v_t$ and for the second $d$ rounds $g_{d+t} = -v_t$. For the remaining rounds, we write $t = i + j\sqrt{d} + 2d$ for $j \in \mathbb{Z}$ and $1 \leq i \leq \sqrt{d}$, and set $g_t = \frac{1}{\sqrt{d}}v_d + \left((-1)^j\sqrt{1 - \frac{1}{d}}\right)v_i$. Intuitively, the losses are cycling with alternating signs through the first $\sqrt{d}$ basis vectors, but always maintain a small positive component in the direction of $v_d$. Notice that since $T - 2d$ is a multiple of $2\sqrt{d}$, the alternating signs imply that $\sum_{t=1}^T g_t$ is a positive scalar multiple of $v_d$. Consider $\mathring{w} = -v_d$. Then, we have:

$$\|\mathring{w}\|_2 \sqrt{\sum_{t=1}^T \|g_t\|_2^2} = O(\sqrt{T})$$

$$\sqrt{\mathrm{rank}(G_T) \sum_{t=1}^T \langle g_t, w_t\rangle^2} = O(\sqrt{T})$$

$$\sqrt{\langle \mathring{w}, G_T^{1/2}\mathring{w}\rangle \mathrm{tr}(G_T^{1/2})} = O\left(\sqrt{T/d^{1/4} + \sqrt{dT}}\right)$$

In this case, the trace of $\sqrt{\sum_{t=1}^T g_t g_t^\top}$ captures the fact that even though the $g_t$ span $d$ dimensions, they are approximately contained in $\sqrt{d}$ dimensions. This allows bound (3) to perform much better than either of the other bounds. In contrast, if the example is modified so that the first $2d$ rounds only cycle between the first $\sqrt{d}$ basis vectors, we would have $\mathrm{rank}(G_T) = \sqrt{d}$ and so the full-matrix bound (2) is the best. Finally, if we increase the component on $v_d$ in each round to, for example, $\frac{1}{\sqrt{2}}$, then the bound (1) is the smallest. Therefore none of the bounds uniformly dominates the others.

To gain a little more intuition for what the bound (3) means, let us investigate the worst-case performance of the bounds (1), (2) and (3) over all $\mathring{w}$ with $\|\mathring{w}\|_2 \leq 1$. To this end, write $T_{\mathrm{eff}} = \sum_{t=1}^T \|g_t\|_2^2$ and let $\lambda_{\max} = \sup_{\|\mathring{w}\| \leq 1} \sum_{t=1}^T \langle g_t, \mathring{w}\rangle^2$. Then we clearly have (1) is $O(\sqrt{T_{\mathrm{eff}}})$ while the bound (2) is at most $O(\sqrt{r\lambda_{\max}})$. On the other hand, by Cauchy-Schwarz inequality we have $\mathrm{tr}(G_T^{1/2}) \leq \sqrt{r_{\mathrm{eff}} T_{\mathrm{eff}}}$ where $r_{\mathrm{eff}} \leq r$ is some "effective rank" that might be much lower than the true rank $r$. With this notation, we have that the bound (3) is at most $(\lambda_{\max} r_{\mathrm{eff}} T_{\mathrm{eff}})^{1/4}$. Thus, we see that the new bound is at most the geometric mean of the bounds (1) and (2), but could potentially be much lower if the effective rank $r_{\mathrm{eff}}$ is smaller than $r$.

## 5.1 Achieving the Optimal Full-Matrix AdaGrad Bound

Now that we see there is some potential advantage to a bound like (3), we will show how to obtain the bound without manually tuning $\eta$ using our framework. The approach is very similar to how we obtained the bound (2): we run Algorithm 2 and in round $t$ we set $\|\cdot\|_t = \|\cdot\|_{G_t^{1/2}}$. With this setting, the desired bound is an almost immediate consequence of Theorem 5:

**Theorem 7.** *Suppose $W \subset \mathbb{R}^d$ and $g_t$ satisfies $\|g_t\|_{\mathbf{2}} \leq 1$ for all $t$. Let $G_t = \sum_{i=1}^t g_i g_i^\top$. Define $\| \cdot \|_t$ be $\|x\|_t^2 = x^\top (I + G_t)^{1/2} x$. Then the regret of Algorithm 2 using these norms is bounded by:*

$$R_T(\mathring{w}) \leq \tilde{O}\left( \sqrt{(\|\mathring{w}\|_{\mathbf{2}}^2 + \mathring{w}^\top G_T^{1/2} \mathring{w}) tr\left(G_T^{1/2}\right)} \right)$$

*where the $\tilde{O}$ notation hides a logarithmic dependency on $tr\left(G_T^{1/2}\right) \sqrt{\|\mathring{w}\|_{\mathbf{2}}^2 + \mathring{w}^\top G_T^{1/2} \mathring{w}}$.*

This Theorem recovers the desired bound (3) up to log factors. Moreover, it is possible to interpret the operation of the algorithm as in some rough sense "learning the optimal learning rate" required for the original AdaGrad algorithm to achieve this bound.

*Proof.* Observe that since $\|g_t\|_{\mathbf{2}} \leq 1$, we have $\|g_t\|_{t-1,\star} = \|g_t\|_{(I+G_{t-1})^{-1/2}} \leq \|g_t\|_{\mathbf{2}} \leq 1$ so that the hypotheses of Theorem 5 are satisfied. In order to complete the analysis we need only calculate:

$$\sum_{t=1}^T \|g_t\|_{t-1,\star}^2 = \sum_{t=1}^T g_t^\top (I + G_{t-1})^{-1/2} g_t \leq \sum_{t=1}^T g_t^\top G_t^{-1/2} g_t \leq 2\mathrm{tr}(G_T^{1/2})$$

Here, in the first inequality, we mildly abuse notation to indicate the pseudo-inverse of $G_t^{1/2}$ as $G_{t-1}^{-1/2}$. The inequalities then follow from Duchi et al. [2010] Lemmas 9 and 10.

Finally, observe that $(I + G_T)^{1/2} \preceq I + G_T^{1/2}$, and apply Theorem 5 to obtain the result. □

## 6 Conclusion

We have introduced online linear optimization algorithms that achieve new full-matrix bounds. We generalizing prior work to arbitary domains, and we provide an improvement on the full-matrix AdaGrad bound. Our results are consequences of a general method for constructing regret bounds in terms of arbitrary sequences of norms.

Our results raise several interesting open questions. Firstly, our full-matrix regret bound seems to be a factor of $\sqrt{\log(T)}$ worse than the best rate in the unconstrained case, suggesting that there is some room to improve our algorithm or analysis. Second, our present results seem restricted to using FTRL algorithms with regularizers centered at the origin. If instead it was possible for the center to vary of the course of the algorithm, one might hope achieve bounds similar to those obtained by [van Erven and Koolen, 2016] in a more general or more efficient manner. Finally, note that in the *unconstrained* setting, Cutkosky and Sarlos [2019] achieves a bound that appears to dispense with the $\sqrt{r}$ factor in (2). Their bound holds only if the gradients $g_t$ satisfy a certain "non-randomness" condition. Although our bounds hold unconditionally, this raises the interesting question of whether their conditional bound can be achieved in the constrained setting.

## Broader Impact

Our work introduces new algorithms and analysis for generic optimization problems. Our contribution is entirely mathematical, and we do not anticipate it engendering negative ethical concerns.

## Acknowledgments and Disclosure of Funding

Much of this work was performed while employed at Google Research.

## Footnotes

[1]Note that the prior bound for the *diagonal* AdaGrad algorithm is different and can improve over (1).

[2]In Duchi et al. [2010], this version of AdaGrad is called the Primal-Dual update version.

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
