[Supplementary Material]

# A Appendix Organization

This appendix is organized as follows: in Section B, C and D we provide the missing proofs of Theorems 3, 4 and 5. In Section E we provide detailed version of Theorems 6 and 7 containing all constants. In Section F we provide a version of Theorem 2 with all constants for completeness.

# B Proof of Theorem 3

In this section we provide the missing proof of Theorem 3, restated below:

**Lemma 3.** *Let $W$ be a real vector space and $\|\cdot\|_1, \ldots, \|\cdot\|_T$ are an increasing sequence of norms on $W$ such that $\frac{1}{2}\|\cdot\|_t$ is $\sigma$-strongly-convex with respect to $\|\cdot\|_t$. Suppose we run FTRL with regularizers given by (6), and with $g_t$ satisfying $\|g_t\|_{t-1,\star} \le 1$ for all t. Then $\|w_t\|_{t-1} \le 1$ for all t, and for all $\mathring{w}$ with $\|\mathring{w}\|_{T-1} \le 1$, the regret of FTRL is bounded by*

$$R_T(\mathring{w}) \le \frac{1}{\sqrt{\sigma}} \left( \|\mathring{w}\|_{T-1}^2 \sqrt{1 + \sum_{t=1}^{T-1} \|g_t\|_{t-1,\star}^2} + \sqrt{\sum_{t=1}^{T} \|g_t\|_{t-1,\star}^2} \right).$$

*Proof.* To begin, observe that since $\psi_t(w) = \infty$ for $\|w\|_t > 1$, the definition of the FTRL update implies $\|w_{t+1}\|_t \le 1$. So now it remains only to show the regret bound.

By the $\sigma$-strong-convexity of $\frac{1}{2}\|\cdot\|_t^2$, we have that $\psi_t$ is $\sqrt{2\sigma + 2\sigma \sum_{i=1}^{t} \|g_i\|_{i-1,\star}^2}$-strongly convex with respect to $\|\cdot\|_t$. Further, since $\|\cdot\|_t$ is increasing with $t$, $\psi_t$ is increasing as well. Therefore direct application of Theorem 1 yields:

$$R_T(\mathring{w}) \le \psi_{T-1}(\mathring{w}) + \sum_{t=1}^{T} \frac{\|g_t\|_{t-1,\star}^2}{2\sqrt{\sigma + \sigma \sum_{i=1}^{t-1} \|g_i\|_{i-1,\star}^2}}$$

Now we recall the following consequence of concavity of the square root function (see Auer et al. [2002], Duchi et al. [2010] for proofs): for any sequence non-negative numbers $x_1, \ldots, x_T$ we have

$$\sum_{t=1}^{T} \frac{x_t}{\sqrt{\sum_{i=1}^{t} x_t}} \le 2\sqrt{\sum_{t=1}^{T} x_t}$$

Using this observation, and the fact that $\|g_t\|_{t-1,\star} \le 1$, we have

$$\sum_{t=1}^{T} \frac{\|g_t\|_{t-1,\star}^2}{2\sqrt{\sigma + \sigma \sum_{i=1}^{t-1} \|g_i\|_{i-1,\star}^2}} \le \sum_{t=1}^{T} \frac{\|g_t\|_{t-1,\star}^2}{2\sqrt{2\sigma}\sqrt{\sum_{i=1}^{t} \|g_i\|_{i-1,\star}^2}}$$

$$\le \sqrt{\frac{1}{\sigma} \sum_{t=1}^{T} \|g_t\|_{t-1,\star}^2}$$

And now the final bound follows by inserting the definition of $\psi_{T-1}$. $\square$

# C Proof of Theorem 4

In this section we provide the missing proof of Theorem 4, restated below:

**Lemma 4.** *Under the assumptions of Lemma 3, for any $\mathring{w} \in W$ (recall we assume $W$ is an entire vector space in Lemma 3), the regret of Algorithm 1 is bounded by:*

$$R_T(\mathring{w}) \le O\left[ \epsilon + \frac{2\|\mathring{w}\|_{T-1}}{\min(1, \sqrt{\sigma})} \max\left( \sqrt{1 + \sum_{t=1}^{T} \|g_t\|_{t-1,\star}^2 \log\left(1 + \frac{\sum_{t=1}^{T} \|g_t\|_{t-1,\star}^2 \|\mathring{w}\|_{T-1}}{\epsilon}\right)}, \right.\right.$$

$$\left.\left. \log\left(1 + \frac{\sum_{t=1}^{T} \|g_t\|_{t-1,\star}^2 \|\mathring{w}\|_{T-1}}{\epsilon}\right) \right) \right]$$

*Proof.* First, by Lemma 3, we have $\|x_t\|_{t-1} \leq 1$, so that $\langle g_t, x_t \rangle \leq \|g_t\|_{t-1,\star}\|x_t\|_{t-1} \leq \|g_t\|_{t-1,\star} \leq 1$. Next, we use an argument from Cutkosky and Orabona [2018]:

$$\sum_{t=1}^{T}\langle g_t, w_t - \mathring{w}\rangle = \sum_{t=1}^{T}\langle g_t, y_t x_t - \mathring{w}\rangle$$

$$= \sum_{t=1}^{T}\langle g_t, x_t\rangle(y_t - \|\mathring{w}\|_{T-1}) + \|\mathring{w}\|_{T-1}\sum_{t=1}^{T}\langle g_t, x_t - \mathring{w}/\|\mathring{w}\|_{T-1}\rangle$$

$$= R_T^{1D}(\|\mathring{w}\|_{T-1}) + R_T^{FTRL}(\mathring{w}/\|\mathring{w}\|_{T-1})$$

where $R_T^{FTRL}$ is the regret of FTRL. Since $\left\|\frac{\mathring{w}}{\|\mathring{w}\|_{T-1}}\right\|_{T-1} = 1$, Lemma 3 tells us:

$$R_T^{FTRL}(\mathring{w}/\|\mathring{w}\|_{T-1}) \leq \frac{2}{\sqrt{\sigma}}\sqrt{1 + \sum_{t=1}^{T-1}\|g_t\|_{t-1,\star}^2}$$

Now it remains to use the regret bound on $\mathcal{A}$. Observe that $|s_t| \leq \|g_t\|_{t-1,\star} \leq 1$, so we can apply the regret bound of Theorem 2. Specifically, if we pull the constants from Theorem 11, we obtain:

$$R_T(\mathring{w}) \leq \epsilon + 2\|\mathring{w}\|_{T-1}\max\left[\sqrt{\left(3 + 3\sum_{t=1}^{T}\|g_t\|_{t-1,\star}^2\right)\log\left(e + \frac{\|\mathring{w}\|_{T-1}(6 + 11\sum_{t=1}^{T}\|g_t\|_{t-1,\star}^2)}{\epsilon}\right)},\right.$$

$$\left.2\log\left(e + \frac{\|\mathring{w}\|_{T-1}(6 + 11\sum_{t=1}^{T}\|g_t\|_{t,-1\star}^2)}{\epsilon}\right)\right]$$

$$+ \frac{2\|\mathring{w}\|_{T-1}}{\sqrt{\sigma}}\sqrt{1 + \sum_{t=1}^{T-1}\|g_t\|_{t-1,\star}^2}$$

$\square$

# D    Proof of Theorem 5

In this section, we provide the missing proof of Theorem 5, restated below:

**Theorem 5.** *Each output $w_t$ of Algorithm 2 lies in $W$, and the regret for any $\mathring{w} \in W$ is at most:*

$$R_T(\mathring{w}) \leq O\left[\epsilon + \frac{\|\mathring{w}\|_{T-1}}{\min(1,\sqrt{\sigma})}\max\left(\sqrt{1 + \sum_{t=1}^{T}\|g_t\|_{t-1,\star}^2\log\left(1 + \frac{\sum_{t=1}^{T}\|g_t\|_{t-1,\star}^2\|\mathring{w}\|_{T-1}}{\epsilon}\right)},\right.\right.$$

$$\left.\left.\log\left(1 + \frac{\sum_{t=1}^{T}\|g_t\|_{t-1,\star}^2\|\mathring{w}\|_{T-1}}{\epsilon}\right)\right)\right]$$

*Proof.* The proof is nearly identical to that Cutkosky and Orabona [2018] Theorem 3 - we simply observe that none of the steps in their proof required a fixed norm, and reproduce the argument for completeness. From Cutkosky and Orabona [2018] Proposition 1, we have that $S_t$ is convex and Lipschitz with respect to $\|\cdot\|_{t-1}$ for all $t$. Therefore we have $\ell_t$ is also convex and $\|g_t\|_{t-1,\star}$-Lipschitz with respect to $\|\cdot\|_{t-1}$. Therefore we have $\|\hat{g}_t\|_{t-1,\star} \leq \|g_t\|_{t-1,\star}$.

$$\sum_{t=1}^{T}\langle g_t, w_t - \mathring{w}\rangle = \sum_{t=1}^{T}\langle g_t, v_t\rangle + \langle g_t, w_t - v_t\rangle - \langle g_t, \mathring{w}\rangle$$

$$\leq \sum_{t=1}^{T}\langle g_t, v_t\rangle + \|g_t\|_{t-1,\star}\|w_t - v_t\|_{t-1} - \langle g_t, \mathring{w}\rangle$$

$$= 2\sum_{t=1}^{T}\ell_t(v_t) - \ell_t(\mathring{w})$$

$$\leq 2\sum_{t=1}^{T}\langle \hat{g}_t, v_t - \mathring{w}\rangle$$

Now since $\|\hat{g}_t\|_{t-1,\star} \leq \|g_t\|_{t-1,\star} \leq 1$, we have that $\sum_{t=1}^{T} \langle \hat{g}_t, v_t - \mathring{w} \rangle$ is simply the regret of the unconstrained Algorithm 1 and so the Theorem follows. Specifically, if we again substitute in the result of Theorem 11 to get all constants, we obtain:

$$R_T(\mathring{w}) \leq \epsilon + 2\|\mathring{w}\|_{T-1} \max \left[ \sqrt{\left( 3 + 3\sum_{t=1}^{T} \|g_t\|_{t-1,\star}^2 \right) \log \left( e + \frac{\|\mathring{w}\|_{T-1}(6 + 11\sum_{t=1}^{T} \|g_t\|_{t-1,\star}^2)}{\epsilon} \right)}, \right.$$

$$\left. 2\log \left( e + \frac{\|\mathring{w}\|_{T-1}(6 + 11\sum_{t=1}^{T} \|g_t\|_{t,-1\star}^2)}{\epsilon} \right) \right]$$

$$+ \frac{2\|\mathring{w}\|_{T-1}}{\sqrt{\sigma}} \sqrt{1 + \sum_{t=1}^{T-1} \|g_t\|_{t-1,\star}^2}$$

$\square$

# E    Detailed Full-Matrix Bounds with Constants

In this section, we show a more detailed proof of Theorems 6 and 7 that includes all constant factors and logarithmic terms fetched from Theorem 11.

First, we prove the following result that was used in the proof of Theorem 6:

**Lemma 8.** *Suppose* $\| \cdot \|_1$ *and* $\| \cdot \|_2$ *are such that* $\frac{1}{2}\|x\|_i^2$ *is* $\sigma_i$*-strongly convex with respect to* $\| \cdot \|_i$ *for* $i \in \{1, 2\}$. *Then the* $\|x\| = \sqrt{\|x\|_1^2 + \|x\|_2^2}$ *is a seminorm and is* $\min(\sigma_1, \sigma_2)$*-strongly convex with respect to* $\| \cdot \|$.

*Proof.* First, we show that $\| \cdot \|$ is a seminorm. It is clear that $\|0\| = 0$ and $c\|x\| = \|cx\|$. To check triangle inequality, we have

$$\|x + y\| = \sqrt{\|x + y\|_1^2 + \|x + y\|_2^2}$$
$$\leq \sqrt{(\|x\|_1 + \|y\|_1)^2 + (\|x\|_2 + \|y\|_2)^2}$$
$$= \|(\|x\|_1, \|x\|_2) + (\|y\|_1, \|y\|_2)\|_{\mathbf{2}}$$
$$\leq \|(\|x\|_1, \|x\|_2)\|_{\mathbf{2}} + \|(\|y\|_1, \|y\|_2)\|_{\mathbf{2}}$$
$$= \|x\| + \|y\|$$

Now we show the strong-convexity. Recall that a function $f$ is $\sigma$-strongly convex if and only if for all $p \in [0, 1]$ and all $x, y$,

$$f(px + (1-p)y) \leq pf(x) + (1-p)f(y) - \frac{\sigma p(1-p)}{2}\|x - y\|^2$$

Let $\sigma = \min(\sigma_1, \sigma_2)$. Then we have

$$\frac{1}{2}\|px + (1-p)y\|_1^2 \leq \frac{p}{2}\|x\|_1^2 + \frac{1-p}{2}\|y\|_1^2 + \frac{\sigma p(1-p)}{2}\|x - y\|_1^2$$
$$\frac{1}{2}\|px + (1-p)y\|_2^2 \leq \frac{p}{2}\|x\|_2^2 + \frac{1-p}{2}\|y\|_2^2 + \frac{\sigma p(1-p)}{2}\|x - y\|_2^2$$

Adding these two inequalities proves the stated strong-convexity.     $\square$

**Theorem 9.** *Suppose* $g_t$ *satisfies* $\|g_t\| \leq 1$ *for all t where* $\| \cdot \|$ *is a norm such that* $\frac{1}{2}\| \cdot \|^2$ *is* $\sigma$*-strongly convex with respect to* $\| \cdot \|$. *Let* $G_t = \sum_{i=1}^{t} g_i g_i^{\top}$ *and let r be the rank of* $G_T$. *Suppose we run Algorithm 2 with* $\|x\|_t^2 = \|x\|^2 + x^{\top}(I + G_t)x$, *where I is the identity matrix. Then we obtain regret:*

$$R_T(\mathring{w}) \leq \epsilon + 2\|\mathring{w}\|_T \max \left[ \sqrt{(3 + 3r\log(T+1)) \log \left( e + \frac{\|\mathring{w}\|_T(7 + 4r\log(T+1))}{\epsilon} \right)}, \right.$$

$$\left. 2\log \left( e + \frac{\|\mathring{w}\|_T(7 + 4r\log(T+1))}{\epsilon} \right) \right] + \frac{2}{\sqrt{\min(\sigma, 1)}}\|\mathring{w}\|_T \sqrt{1 + r\log(T+1)}$$

*Proof.* We saw in the proof of Theorem 6 that $\|\mathring{w}\|_{T-1} \leq \|w\|_T = \sqrt{2\|w\|_{\mathbf{2}}^2 + \sum_{t=1}^{T} \langle g_t, \mathring{w} \rangle^2}$. We also saw:

$$\sum_{t=1}^{T} \|g_t\|_{t-1,\star}^2 \leq \text{rank}(G_T) \log(T+1)$$

So then with all constants, the regret is

$$R_T(\mathring{w}) \le \epsilon + 2\|\mathring{w}\|_{T-1} \max\left[ \sqrt{\left(3 + 3\sum_{t=1}^{T} \|g_t\|_{t-1,\star}^2\right) \log\left(e + \frac{\|\mathring{w}\|_{T-1}(7 + 4\sum_{t=1}^{T} \|g_t\|_{t-1,\star}^2)}{\epsilon}\right)}, \right.$$

$$\left. 2\log\left(e + \frac{\|\mathring{w}\|_T(7 + 4\sum_{t=1}^{T} \|g_t\|_{t-1,\star}^2)}{\epsilon}\right) \right]$$

$$+ \frac{2}{\sqrt{\min(\sigma, 1)}} \sqrt{1 + \sum_{t=1}^{T-1} \|g_t\|_{t-1,\star}^2}$$

$$\le \epsilon + 2\|\mathring{w}\|_T \max\left[ \sqrt{(3 + 3r\log(T+1))\log\left(e + \frac{\|\mathring{w}\|_T(7 + 4r\log(T+1))}{\epsilon}\right)}, \right.$$

$$\left. 2\log\left(e + \frac{\|\mathring{w}\|_T(7 + 4r\log(T+1))}{\epsilon}\right) \right]$$

$$+ \frac{2}{\sqrt{\min(\sigma, 1)}} \|\mathring{w}\|_T \sqrt{1 + r\log(T+1)}$$

$$\square$$

Next, we carry out a similar computation for the AdaGrad-style full-matrix algorithm:

**Theorem 10.** *Suppose $W \subset \mathbb{R}^d$ and $g_t$ satisfies $\|g_t\|_2 \le 1$ for all $t$. Let $G_t = \sum_{i=1}^{t} g_i g_i^\top$. Define $\|\cdot\|_t$ be $\|x\|_t^2 = x^\top (I + G_t)^{1/2} x$. Then the regret of Algorithm 2 using these norms is bounded by:*

$$R_T(\mathring{w}) \le \tilde{\epsilon} + 2\|\mathring{w}\|_T \max\left[ \sqrt{\left(3 + 6tr(G_T^{1/2})\right)\log\left(e + \frac{\|\mathring{w}\|_T(7 + 8tr(G_T^{1/2}))}{\epsilon}\right)}, \right.$$

$$\left. 2\log\left(e + \frac{\|\mathring{w}\|_T(7 + 8tr(G_T^{1/2}))}{\epsilon}\right) \right] + 2\|\mathring{w}\|_T \sqrt{1 + 2tr(G_T^{1/2})}$$

*where the $\tilde{O}$ notation hides a logarithmic dependency on $tr\left(G_T^{1/2}\right)\sqrt{\|\mathring{w}\|_2^2 + \mathring{w}^\top G_T^{1/2} \mathring{w}}$.*

This Theorem recovers the desired bound (3) up to log factors. Moreover, it is possible to interpret the operation of the algorithm as in some rough sense "learning the optimal learning rate" required for the original AdaGrad algorithm to achieve this bound.

*Proof.* In the proof of Theorem 7, we saw $\|\mathring{w}\|_{T-1} \le \|\mathring{w}\|_T = \sqrt{\|\mathring{w}\|_2^2 + \mathring{w}^\top G_T^{1/2} \mathring{w}}$. Further,

$$\sum_{t=1}^{T} \|g_t\|_{t-1,\star}^2 \le 2tr(G_T^{1/2})$$

So then with all constants, the regret is

$$R_T(\mathring{w}) \le \epsilon + 2\|\mathring{w}\|_{T-1} \max\left[ \sqrt{\left(3 + 3\sum_{t=1}^{T} \|g_t\|_{t-1,\star}^2\right) \log\left(e + \frac{\|\mathring{w}\|_{T-1}(7 + 4\sum_{t=1}^{T} \|g_t\|_{t-1,\star}^2)}{\epsilon}\right)}, \right.$$

$$\left. 2\log\left(e + \frac{\|\mathring{w}\|_T(7 + 4\sum_{t=1}^{T} \|g_t\|_{t-1,\star}^2)}{\epsilon}\right) \right]$$

$$+ \frac{2}{\sqrt{\sigma}} \sqrt{1 + \sum_{t=1}^{T-1} \|g_t\|_{t-1,\star}^2}$$

$$\le \epsilon + 2\|\mathring{w}\|_T \max\left[ \sqrt{\left(3 + 6tr(G_T^{1/2})\right)\log\left(e + \frac{\|\mathring{w}\|_T(7 + 8tr(G_T^{1/2}))}{\epsilon}\right)}, \right.$$

$$\left. 2\log\left(e + \frac{\|\mathring{w}\|_T(7 + 8tr(G_T^{1/2}))}{\epsilon}\right) \right] + 2\|\mathring{w}\|_T \sqrt{1 + 2tr(G_T^{1/2})}$$

$$\square$$

# F   Full Version of Theorem 2 with Constants

In this section, we provide a more detailed version of Theorem 2 including all logarithmic and constant factors. The proof is essentially a (slightly looser) version of analysis in Cutkosky and Sarlos [2019], but we provide it below for completeness.

**Theorem 11.** *There exists a one-dimensional online linear optimization algorithm such that if $|g_t| \le 1$ for all t, the regret is bounded by*

$$\sum_{t=1}^{T} g_t(w_t - \mathring{w}) \le \epsilon + 2|\mathring{w}| \max \left[ \sqrt{\left(3 + 3\sum_{t=1}^{T} g_t^2\right) \log\left(e + \frac{|\mathring{w}|(7 + 4\sum_{t=1}^{T} g_t^2)}{\epsilon}\right)}, \right.$$

$$\left. 2\log\left(e + \frac{|\mathring{w}|(7 + 4\sum_{t=1}^{T} g_t^2)}{\epsilon}\right)\right]$$

*And moreover each $w_t$ is computed in $O(1)$ time.*

*Proof.* Define the *wealth* of an algorithm as:

$$\text{Wealth}_t = \epsilon - \sum_{\tau=1}^{t} g_\tau w_\tau$$

We set

$$w_{t+1} = v_{t+1}\text{Wealth}_t$$

where $v_t \in [-1/2, 1/2]$. This implies:

$$\text{Wealth}_T = \epsilon \prod_{t=1}^{T}(1 - g_t v_t)$$

Define

$$\text{Wealth}_T(\mathring{v}) = \epsilon \prod_{t=1}^{T}(1 - g_t \mathring{v})$$

Now, to choose $v_t$, consider the functions:

$$\ell_t(v) = -\log(1 - g_t v)$$

Observe that $\ell_t(v)$ is convex. Let $z_t = \frac{g_t}{1 - g_t v_t} = \ell'_t(v_t)$. Notice that $|z_t| \le 2|g_t| \le 2$ since $v_t \in [-1/2, 1/2]$. Then we have

$$\log\left(\text{Wealth}_T(\mathring{v})\right) - \log\left(\text{Wealth}_T\right) = \sum_{t=1}^{T} \ell_t(v_t) - \ell_t(\mathring{v}) \le \sum_{t=1}^{T} z_t(v_t - \mathring{v})$$

Now we choose $v_t \in [-1/2, 1/2]$ using FTRL on the losses $z_t$ with regularizers

$$\psi_t(v) = \frac{Z}{2}(5 + \sum_{\tau=1}^{t} z_\tau^2)v^2$$

Notice that $\psi_t$ is $Z(4 + \sum_{\tau=1}^{t} z_\tau^2)$-strongly convex with respect to $|\cdot|$. Therefore by Theorem 1:

$$\sum_{t=1}^{T} z_t(v_t - \mathring{v}) \le \psi_T(\mathring{v}) + \frac{1}{2}\sum_{t=1}^{T} \frac{z_t^2}{Z(5 + \sum_{\tau=1}^{t-1} z_\tau^2)}$$

$$\le \frac{Z}{2}\left(5 + \sum_{t=1}^{T} z_t^2\right)\mathring{v}^2 + \frac{1}{2Z}\sum_{t=1}^{T} \frac{z_t^2}{1 + \sum_{\tau=1}^{t} z_\tau^2}$$

$$\le \frac{Z}{2}\left(5 + \sum_{t=1}^{T} z_t^2\right)\mathring{v}^2 + \frac{1}{2Z}\log\left(1 + \sum_{t=1}^{T} z_t^2\right)$$

Therefore, for all $\mathring{v} \in [-1, 2/, 1/2]$,

$$\log\left(\text{Wealth}_T\right) \ge \log\left(\text{Wealth}_T(\mathring{v})\right) - \frac{Z}{2}\left(5 + \sum_{t=1}^{T} z_t^2\right)\mathring{v}^2 + \frac{1}{2Z}\log\left(1 + \sum_{t=1}^{T} z_t^2\right)$$

$$\ge \log\left(\text{Wealth}_T(\mathring{v})\right) - \frac{Z}{2}\left(5 + 4\sum_{t=1}^{T} g_t^2\right)\mathring{v}^2 + \frac{1}{2Z}\log\left(1 + 4\sum_{t=1}^{T} g_t^2\right)$$

Next, use the tangent bound $\log(1 - x) \geq -x - x^2$ to obtain:

$$\log\left(\text{Wealth}_T(\mathring{v})\right) \geq \log(\epsilon) - \sum_{t=1}^{T} g_t \mathring{v} - \sum_{t=1}^{T} g_t^2 \mathring{v}^2$$

So overall we have:

$$\log\left(\text{Wealth}_T\right) \geq \log(\epsilon) - \sum_{t=1}^{T} g_t \mathring{v} - \frac{Z}{2}\left(5 + \sum_{t=1}^{T}\left(\frac{2}{Z} + 4\right)g_t^2\right)\mathring{v}^2 - \frac{1}{2Z}\log\left(1 + 4\sum_{t=1}^{T} g_t^2\right)$$

$$\text{Wealth}_T \geq \epsilon \exp\left(-\sum_{t=1}^{T} g_t \mathring{v} - \frac{Z}{2}\left(5 + \sum_{t=1}^{T}(\frac{2}{Z} + 4)g_t^2\right)\mathring{v}^2 - \frac{1}{2Z}\log\left(1 + 4\sum_{t=1}^{T} g_t^2\right)\right)$$

Now we relate this to regret:

$$\sum_{t=1}^{T} g_t(w_t - \mathring{w}) = \epsilon - \mathring{w}\sum_{t=1}^{T} g_t - \text{Wealth}_T$$

$$\leq \epsilon - \mathring{w}\sum_{t=1}^{T} g_t - \epsilon \exp\left(-\sum_{t=1}^{T} g_t \mathring{v} - \frac{Z}{2}\left(5 + \sum_{t=1}^{T}\left(\frac{2}{Z} + 4\right)g_t^2\right)\mathring{v}^2 - \frac{1}{2Z}\log\left(1 + 4\sum_{t=1}^{T} g_t^2\right)\right)$$

$$\leq \epsilon + \sup_G\left[G\mathring{w} - \epsilon \exp\left(G\mathring{v} - \frac{Z}{2}\left(5 + \sum_{t=1}^{T}\left(\frac{2}{Z} + 4\right)g_t^2\right)\mathring{v}^2 - \frac{1}{2Z}\log\left(1 + 4\sum_{t=1}^{T} g_t^2\right)\right)\right]$$

$$\leq \epsilon + \frac{|\mathring{w}|}{\mathring{v}}\left(\log\left(\frac{|\mathring{w}|}{\epsilon\mathring{v}}\right) + \frac{Z}{2}\left(5 + \sum_{t=1}^{T}\left(\frac{2}{Z} + 4\right)g_t^2\right)\mathring{v}^2 + \frac{1}{2Z}\log\left(1 + 4\sum_{t=1}^{T} g_t^2\right) - 1\right)$$

$$\leq \epsilon + \frac{|\mathring{w}|}{\mathring{v}}\log\left(\frac{|\mathring{w}|(1 + 4\sum_{t=1}^{T} g_t^2)^{1/2Z}}{\epsilon\mathring{v}}\right) + \frac{Z}{2}\left(5 + \sum_{t=1}^{T}\left(\frac{2}{Z} + 4\right)g_t^2\right)\mathring{v}$$

where we have used Cutkosky and Sarlos [2019] Lemma 3 in to calculate the supremum over $G$. Now set $Z = 1$, apply Cutkosky and Sarlos [2019] Lemma 4, and over-approximate several constants to obtain:

$$\sum_{t=1}^{T} g_t(w_t - \mathring{w}) \leq \epsilon + 2|\mathring{w}| \max\left[\sqrt{\left(3 + 3\sum_{t=1}^{T} g_t^2\right)\log\left(e + \frac{|\mathring{w}|(7 + 4\sum_{t=1}^{T} g_t^2)}{\epsilon}\right)},\right.$$

$$\left. 2\log\left(e + \frac{|\mathring{w}|(7 + 4\sum_{t=1}^{T} g_t^2)}{\epsilon}\right)\right]$$

$\square$