[Reviews · NeurIPS 2020]

Review 1

Summary and Contributions: This paper deals with the setting of parameter-free online convex optimization. In particular, "full-matrix" bounds are considered. These bounds take into account the fact that the rank of the subspace spanned by the subgradients received along the game can be smaller than the true dimension d. Importantly, the authors provide 3 different characterisations of matrix bounds which capture different aspects of the problem at hand. At the same time, they describe algorithms achieving the bounds given in the introduction.

Strengths: This work follows the recent developments in parameter-free online learning. Given the various applications of these algorithms in practice, it is fundamental to understand how to better design them. This work provides a useful approach, both to design new algorithms and shed light on already existing ones (such as Adagrad). Compared to prior work, the algorithms provided by the authors do not need prior knowledge of the norm of the (optimal) competitor and their regret bounds hold for any arbitrary convex domain. The paper is well written: the problem is nicely introduced and clearly explained. Also, the technical level is satisfactory and all the theorems are carefully explained.

Weaknesses: In my opinion, the techniques used in this work are all somehow previously known, as stated by the authors themselves. For example, Algorithm 1 is an application of the scheme proposed in [17] by Cutkosky & Orabona on how to combine online learning algorithms and the proof of its regret bound seems straightforward (both the regret of FTRL and 1d parameter-free optimization algorithms are well known). Also, Algorithm 2 requires a straightforward extension of Thm 3 from [17] (as honestly stated also by the authors). The authors also provide a brief discussion about the result of Theorem 6 compared to similar results contained in [17], however it is not totally clear to me why the algorithm in [17] cannot be applied to constrained settings. Also, I'm not an expert on this but I think from a practical point of view the diagonal version of Adagrad is used and not the full matrix version, which requires dealing with a dxd matrix in each round, which makes the running time much worse. Regarding this, I think an empirical evaluation of the algorithms would be beneficial. For example, it would be nice to see if the theoretical gains of the algorithm proposed in Theorem 7 are justified despite a worse running time. More in general, a it would be nice to show an empirical evaluation against the original Adagrad algorithm but also to other parameter-free methods such as the ones given in [23] (Cutkosky and Sarlos). **After rebuttal**: the authors clearly explained why the reduction in [17] does not work in this case in their response. However, this is not clearly stated in the main paper and I think it should. Nevertheless, the paper is relevant and well written.

Correctness: All the theorems are sound.

Clarity: The paper is very well written, easy to follow and enjoyable to read. I think there are some typos in the proof of Theorem 6. In particular, the authors say (line 155) that " $ \| x \|_t^2 = \| x \|^2 + x^\top ( I + G_T ) x $". However, at the beginning of the proofs (line 157) we have that: \| x \|^2 = \| \mathring{x} \|^2 x^\top ( I + G_T ) x = \| x \| + ... ", which is different from the definition given above (while after the equality a square is missing from \| x \| ). Also, in the proof of the same theorem, lemma 8 is used but its statement can't be found in the main paper (I think it should be mentioned it is in the Appendix).

Relation to Prior Work: In my opinion this work only looks at one side of the problem, which is the norm of the competitor. On the other hand, from a practical point of view it is of equal importance (if not more important) to consider also the norms of the gradients. In this work it is assumed they’re upper bounded by 1. However, a major reason for adagrad success can be the fact that it is a scale-free algorithm (see for example “Scale-Free Online Learning” [Orabona & Pal, 2018]), meaning that if the gradients are multiplied by different constants, then the learning rates will scale them. This is crucial with neural-networks, where the first layers have smaller magnitude of the gradients compared to the last layers. I am aware of the fact that it's not possible to achieve an optimal dependence on both the norm of the competitor and an upper bound on the norm of the gradients without prior knowledge, however I would like to see a discussion on the aforementioned behaviour.

Reproducibility: Yes

Additional Feedback:


Review 2

Summary and Contributions: This paper considers online convex optimization by automatically tuning the learning rate to an optimal value. They devise an algorithm, based on FTRL framework, to attain a regret bound (4) depend on the sequence of the given norms. As the application of this framework, they recover various full-matrix bounds up to a factor log T.

Strengths: They provide a vehicle for designing parameter-free algorithms that is adaptive to the series of given norms. This framework can recover several prior results up to a logarithmic factor by the proper substitutions of norms. I believe it is insightful for designing parameter-free algorithms for more complicated tasks. Also, their regret bound is specific to the sequence of gradients (or subgradients) and the optimal solution.

Weaknesses: (1) It is a mystery why they need to pay an extra log T multiplicative factor for the regret bounds. As the author mentioned, there might be something missing in the analysis or algorithm. (2) For section 3.2, the author applies the technique in [17] to derive regret bound when the domain is constrained. However, this regret loses the information of the domain. It is better if the regret can also be specific to the domain.

Correctness: The paper is based on the sharp observations from the previous works. The analysis is simple and solid.

Clarity: One thing unclear is about the assumptions of Theorem 5. It follows from Lemma 4 and 3, the author need to make clear if the sequence of norms should be increasing or not. Also, Line 136 claims the algorithm 2 can proceeds with arbitrary norms. I hope it can be clear that if the increase of the sequence of norms is necessary.

Relation to Prior Work: Clear and concise.

Reproducibility: Yes

Additional Feedback: Since the algorithm 2 wraps algorithm 1 and Diagonal betting algorithm [23] as the ingredients. I am a little worried about the efficient issue. Could the author compare the computational complexity of algorithm 2 with other paprameter-free algorithms? __________________________________________After rebuttal______________ I am happy with the author's response. I would like to increase one more score to encourage this work.


Review 3

Summary and Contributions: The paper studies the setting on convex optimization where the decision set can be either constrained and unconstrained and presents algorithms with improved regret bounds compared to AdaGrad.

Strengths: What I like about this paper is the idea of delegating the parameter tuning task that is usually required for FTRL to a parameter-free algorithm using the dimension-free-to-one-dimensional reduction by Cutkosky and Orabona 2018. This idea showcases nicely how FTRL and parameter-free algorithms can be combined to yield an *adaptive* parameter-free regret bound. To achieve the required adaptivity, the authors introduce an intermediate algorithm that uses varying norms, with an interesting analysis.

Weaknesses: The paper's main (advertised) contributions are algorithms that achieve the regret bounds in Eq. (2) and (3). For the former bound, the authors only claim to be the first to achieve it for the constrained setting. Unfortunately, an algorithm with a tighter regret bound than (2) already exists (in both the constrained and unconstrained settings). In fact, this is achieved by the Matrix-FreeGrad algorithm due to Mhammedi and Koolen 2020 'Lipschitz and Comparator-Norm Adaptivity in Online Learning'; Inspecting the display immediately after their Theorem 10, one can see that the regret bound of Matrix-FreeGrad is essentially (in the notation of the current manuscript) |w|_{G_T} \sqrt{\log \det (I + G_T)} which is always smaller than Eq. (2) (this follows from the steps between lines 163 and 164 of the current manuscript). The regret bound of Matrix-Freegrad can be seamlessly transferred to the constrained setting via the constrained-to-unconstrained reduction due to Cutkosky and Orabona 2018 or Cutkosky 2020 'Parameter-free, Dynamic, and Strongly-Adaptive Online Learning'. The authors claim that an algorithm with regret bound (2) in the unconstrained setting already exists and refer to the paper by Cutkosky and Orabona 2018. I do not think that the latter paper achieves this bound with r equal to G_T's rank (r is equal to the full dimension in their case). That being said, if there were an algorithm that achieves such a bound in the unconstrained setting (such as Matrix-FreeGrad), then the constrained-to-unconstrained reduction due to Cutkosky and Orabona 2018 (or Cutkosky 2020 'Parameter-free, Dynamic, and Strongly-Adaptive Online Learning') will immediately lead to the same regret-bound in the constrained setting (up to lower-order terms). I actually do not see why Algorithm 2 (Varying Norm Adaptivity) is needed to transfer a regret bound to the constrained setting. The black box reduction due to Cutkosky and Orabona 2018 does not care about the inner workings of the unconstrained algorithm to transfer the regret bound, as long as the norm of the gradients is bounded by 1 (for some fixed choice of norm). Perhaps Algorithm 2 relaxes the requirement on the norm of the gradients; only requiring |g_t|_{t-1,*}\leq 1 instead of the more constraining |g_t|_* \leq 1. However, this is never discussed in the paper. Also, in lines 53 and 54, the authors claim that prior work has only considered fixed norms. This is not true; see the work of Cutkosky 2019 'Combining Online Learning Guarantees.' On minor aspects: - line 63, a square-root is missing in the definition of the |x|_M. - Line 67, the subgradient should be defined. - Line 114, there is a missing subscript 't' in the norm. - Lemma 4 does not mention where the comparator w lives---it should be unconstrained. =========== Post rebuttal ============ It seems that Eq (2) does not trivially follow from the reduction by Cutkosky and Orabona. The discussion provided in the authors' response is highly relevant to the paper, as it motivates the problem and shows where previous methods fail. I do not understand why this was not included in the manuscript in the first place. For this reason, I only increase my score to 6, and I would urge the authors to include the explanation provided in the rebuttal in the final version of the paper.

Correctness: Some false claims were made about the novelty of the regret bound in Eq. (2) and the difficulty to transfer unconstrained regret bounds to the constrained setting (see weaknesses section above for more detail).

Clarity: The paper is clear enough.

Relation to Prior Work: Relevant related work missing such as: - Cutkosky 2019 'Combining Online Learning Guarantees'. - Cutkosky 2019 'Artificial Constraints and Hints for Unbounded Online Learning'. - Cutkosky 2020 'Parameter-free, Dynamic, and Strongly-Adaptive Online Learning'. - Mhammedi and Koolen 'Lipschitz and Comparator-Norm Adaptivity in Online Learning'.

Reproducibility: Yes

Additional Feedback: I think the paper would be stronger if it was more focused on the idea of delegating the tuning of the learning rate of FTRL to a parameter-free algorithm. It seems that the algorithm of Thm 6 admits a better regret bound which matches that of Matrix-FreeGrad, but the techniques used to get it in the current paper are rather different. This could be investigated more and discussed in the paper. Finally, I think that the authors should not claim Eq. (2) is novel in the constrained setting.

[Author Response · NeurIPS 2020]

Many thanks to all the reviewers for their time and attention to our work!

**For Reviewer 1:** We certainly agree that our techniques are modifications of prior ideas, but we feel that significant
insight was required to develop and apply them in our particular manner. This is hard to prove as, once written down,
the approach is quite simple (although this is of course actually a positive trait). A difficulty here is that one might be
tempted to believe that at least our Theorem 6 can be easily obtained directly from prior results, and it is only after
trying a bit (as we did for quite some time) that one concludes that things are not so easy. For some more evidence in
this regard, and for your question about applying [17] in constrained settings, please see our response to Reviewer 3 on
how the naive approach can fail. We'd also like to stress that obtaining Theorem 7 is perhaps a better exemplar since
there isn't even an obvious-but-secretly-problematic approach until viewed under our lens.

We also agree it would be better to be scale-free as well as parameter-free, but as you state this is not possible in general.
In fact, it seems even the compromise of adding a penalty of $O(\|\mathring{w}\|^3 + \sqrt{T})$ cannot be achieved without sacrificing the
second-order gradient statistics in the bounds [Mhammedi&Koolen 2020]. That said, we agree that finding what is
achievable here is a good problem - in particular we suspect this may be possible in the constrained setting subject to an
additive penalty of $O(GD)$, where $G$ is the maximum Lipschitz constant and $D$ is the diameter of the domain.

**For Reviewer 2:** Regarding efficiency: The parameter-free scaling algorithm can run in $O(d)$ time per update as it
just needs the inner-product $\langle g_t, x_t \rangle$ to compute its loss. The FTRL algorithm requires the same matrix manipulations
that a corresponding unconstrained algorithm would require to get the same regret bound, and so for the full-matrix
algorithms here this involves a $O(d^2)$ rank-one update step, and the final combined algorithm requires a projection step
that may depend on the domain. Concretely, the algorithm of Theorem 7 runs as fast as full-matrix AdaGrad.

**For Reviewer 3:** We appreciate your comments, and the references you suggest are highly relevant - in particular the
recent FreeGrad algorithm of [Mhammedi&Koolen 2020] is a better comparison in the unconstrained case than the
full-matrix algorithm of [17].

In regards to novelty, we must disagree here. It is certainly very reasonable to suspect that the unconstrained-to-
constrained technique suggested by [17] can be used to immediately convert an algorithm like full-matrix FreeGrad into
a constrained algorithm - we also thought this would work at first! **However, it does not**. As it turns out, the fact that
this approach is not as easy as it seems was actually our original motivation for working on this problem!

Using the constraint-set reduction of [17] in concert with an unconstrained algorithm like FreeGrad is problematic
because the reduction changes the losses supplied to FreeGrad, which will in turn change its regret guarantee. This is
because the regret bound of FreeGrad depends in a delicate manner on the values of the $g_t$. If the original losses are $g_t$,
and the gradients supplied to FreeGrad are $\tilde{g}_t$, the final regret will be $\tilde{O}\left(\sqrt{\tilde{r}\sum_{t=1}^{T}\langle\tilde{g}_t, \mathring{w}\rangle^2}\right)$, where $\tilde{r}$ is the rank of
the $\tilde{g}_t$. It is not clear what the relationship is between this quantity and the desired bound $\tilde{O}\left(\sqrt{r\sum_{t=1}^{T}\langle g_t, \mathring{w}\rangle^2}\right)$.

Here is a sketch of what one might try, and where it goes wrong. To start, we need to pick a norm to use with the
constraint set reduction. The natural choice here is $\|\cdot\|_{G_T}$, so let us say we already know the matrix $G_T$ ahead of time,
and use this norm so that we have $\|\tilde{g}_t\|_{G_T^{-1}} \leq \|g_t\|_{G_T^{-1}}$. This is unrealistic, but even with this extra power it is not clear
that things work: using the most obvious algebraic path of bounding $\sum_{t=1}^{T}\langle\mathring{w}, \tilde{g}_t\rangle^2 \leq \sum_{t=1}^{T}\|\mathring{w}\|_{G_T}^2\|\tilde{g}_t\|_{G_T^{-1}}^2$ yields
a bound like $\tilde{O}\left(\|\mathring{w}\|_{G_T}\sqrt{r\sum_{t=1}^{T}\|g_t\|_{G_T^{-1}}^2}\right)$. Now, the $\sum_{t=1}^{T}\|g_t\|_{G_T^{-1}}^2$ term may add an extra dependence on $r$ that
prevents us from achieving the desired bound. To make this work, we need to guarantee that the matrix $\sum_{t=1}^{T}g_t g_t^{\top}$
induces the same norm up to an absolute constant as the matrix $\sum_{t=1}^{T}\tilde{g}_t\tilde{g}_t^{\top}$, and it is not clear that this will hold. We
are aware that for the case of Metagrad-style bounds [van Erven&Koolen 2016], the work of [Mhammedi et. al 2019]
invokes some clever algebra to show that the reduction works as-is, but their technique does not seem to apply to get the
bounds we are looking for. In contrast, our approach does not encounter these issues, and so we feel justified in saying
that we are in fact the first to provide an algorithm achieving the desired full-matrix bound in the constrained setting.

Moreover, even if there were some unknown algebraic trick that would make the prior reduction apply, **our approach
has obvious additional benefits**: we can easily obtain the optimally-tuned full-matrix AdaGrad-style bound that is not
clear how to obtain (even in the unconstrained case) using other techniques.

We hope this addresses your concern.

[Meta-Review · NeurIPS 2020]

The reviewers were convinced that the reduction from Cutkosky and Orabona, 2018 does not solve the problem already, and agreed on the theoretical contribution of this work. We believe that the paper will benefit from a revision with the following two improvements: 1) include detailed discussions related to Cutkosky and Orabona, 2018 as mentioned in the rebuttal; 2) conduct experiments to showcase the practical advantages of the proposed algorithms compared to existing ones such as AdaGrad, MetaGrad [1], and those from [2,3]. Indeed, one of the key motivations of the paper is to explain the practical success of AdaGrad and to develop an even better parameter-free version, and the results would have been much more convincing if some experimental evidence was included as well. [1] Tim van Erven and Wouter M. Koolen. Metagrad: Multiple learning rates in online learning. 2016. [2] Haipeng Luo, Alekh Agarwal, Nicolò Cesa-Bianchi and John Langford. Efficient Second Order Online Learning via Sketching. 2016. [3] Zakaria Mhammedi and Wouter M. Koolen. Lipschitz and Comparator-Norm Adaptivity in Online Learning. 2020.